# Towards Highly Efficient Cesium Titanium Halide Based Lead-Free Double Perovskites Solar Cell by Optimizing the Interface Layers

**DOI:** 10.3390/nano12193435

**Published:** 2022-09-30

**Authors:** Syed Abdul Moiz, Saud Abdulaziz Albadwani, Mohammed Saleh Alshaikh

**Affiliations:** Device Simulation Laboratory, Department of Electrical Engineering, College of Engineering and Islamic Architecture, Umm Al-Qura University, Makkah 21955, Saudi Arabia

**Keywords:** perovskite solar cell, lead-free perovskite, *Cs_2_TiBr_6_*, *Cs_2_TiI_6_*, *Cs_2_TiCl_6_*, *PEDOT:PSS*, *Nb_2_O_5_*, SCAPS-1D

## Abstract

Lead halide perovskites are the most promising compared to the other recently discovered photovoltaic materials, but despite their enormous potential, these materials are facing some serious concerns regarding lead-based toxicity. Among many lead-free perovskites, the vacancy-ordered double perovskite cesium titanium halide family (*Cs_2_TiX_6_*, *X* = *Cl*, *Br*, *I*) is very popular and heavily investigated and reported on. The main objective of this study is to design and compare an efficient cesium titanium halide-based solar cell that can be used as an alternative to lead-based perovskite solar cells. For efficient photovoltaic requirements, the hole-transport layer and electron-transport layer materials such as PEDOT:PSS and Nb_2_O_5_ are selected, as these are the commonly reported materials and electronically compatible with the cesium titanium halide family. For the active layer, cesium titanium halide family members such as *Cs_2_TiCl_6_*, *Cs_2_TiBr_6_*, and *Cs_2_TiI_6_* are reported here for the devices *ITO/Nb_2_O_5_/Cs_2_TiI_6_/PEDOT:PSS/Au*, *ITO/Nb_2_O_5_/Cs_2_TiBr_6_/PEDOT:PSS/Au*, and *ITO/Nb_2_O_5_/Cs_2_TiCl_6_/PEDOT:PSS/Au*, respectively. To determine the most efficient photovoltaic response, all the layers (*PEDOT:PSS*, *Nb_2_O_5_*, and active perovskite layer) of each device are optimized concerning thickness as well as doping density, and then each optimized device was systematically investigated for its photovoltaic responses through simulation and modeling. It is observed that the device *ITO/Nb_2_O_5_/Cs_2_TiI_6_/PEDOT:PS/Au* shows the most efficient photovoltaic response with little above 18.5% for maximum power-conversion efficiency.

## 1. Introduction

Energy is one of the most important parameters to sustain the current civilization and keep the pace of industrial, social, economic, and commercial growth and developments for the next generation [1,2]. Current sources of conventional fossil energy reserves are being sharply exhausted at an alarming rate, and it is unanimously accepted that these conventional resources are not enough to fulfill the future energy demand. Thus, synergetic efforts are being carried out by various research groups to discover (i) unlimited, (ii) environmentally friendly, (iii) low cost, (iv) easy, and (v) globally available renewable energy sources [3,4,5]. Among these renewable resources, energy from solar photovoltaic (PV) cells is considered one of the best choices, which has the full potential to accomplish all the above requirements [6,7]. Until now, single crystalline Si solar cells show the best performance for power-conversion efficiency and long-term stability [8]. However, the cost associated with single-crystalline Si PV cells is considerably high compared to the other conventional energy resources [9]. Hence, various types of photovoltaic materials that deal with low-cost and high photovoltaic responses have been extensively explored in recent years.

Hybrid organic-inorganic perovskites are the most impressive emerging materials which offer exceptionally high electron-hole diffusion lengths, room temperature device fabrication process, solution processing capabilities, and excellent optical absorption coefficients [10], which are well suitable for low-cost solar cells with superior power-conversion efficiency. Therefore, the power-conversion efficiency of perovskite solar cells has jumped from 1% to more than 28% within very few years of research and development [11]. It is unprecedented and represents remarkable progress shown by any type of solar cell. Nevertheless, the best-reported material used as an active perovskite layer for efficient photovoltaic response is lead-based perovskite compounds such as methylammonium lead trihalide (*CH_3_NH_3_PbX_3_*), where lead (*Pb*) is a toxic material in nature and its unreasonable exposure can cause serious consequences to the environment. The scientific community believes that lead-based toxicity presents some limitations for their commercial applications in the future [12,13,14,15]. Therefore, lead-free perovskites with excellent photovoltaic properties are essentially required for next-generation perovskite solar cells.

On the other hand, most of the reported lead-free perovskites are not very efficient compared to methylammonium lead trihalide [16,17]. Recently, the cesium titanium halide family is gaining popularity as lead-free perovskite, where titanium is used as an alternative to replacing lead metal. Unlike lead, titanium is a nontoxic and robust material; it is highly bio-compatible to the environment and abundantly available. Recently, Ju and Chen [18,19,20] reported a novel family of titanium-based perovskites such as *Cs_2_TiX_6_*, where *X* can be a member of the halide group as chlorine (*Cl*), bromine (*Br*), or iodine (*I*). These perovskite materials inherently present tetravalent cations with vacancies, which in turn offer a double perovskite structure. It is observed that the double perovskite structure helps not only for good mechanical and thermal stability but also a tunable energy bandgap, as well as excellent optical absorption. Owing to these advantages, the *Cs_2_TiX_6_* family is emerging as a possible candidate to replace lead-based perovskite solar cells [18,19,20]. Despite early works, the existing study of the *Cs_2_TiX_6_* family is still not completed, and there is a huge research gap for further investigations. Therefore, three distinct family members, namely *Cs_2_TiI_6_*, *Cs_2_TiBr_6_*, and *Cs_2_TiCl_6_*, are selected for this study.

Simulation and modeling of solar cells have already been practiced since the initial stage when limited computer software resources and skilled manpower were available [21,22,23,24]. Now, it is a highly developed and well-established research field, where excellent computational software and hardware facilities are available, and, therefore, it has become the backbone of the recent photovoltaics industry. Modern simulation and modeling techniques for various photovoltaic devices depend on some coupled semiconductor differential equations, which are comprehensively modeled and rigorously tested with experimental results for various types of solar cells, and now they are universally accepted models for photovoltaic devices. These semiconductor differential equations are governed by various geometrical parameters of the device as well as different physical, optical, and electrical parameters of the constituent materials to define the overall photovoltaic behavior of the given solar cell. In scientific literature, various types of simulation software are testified to model the photovoltaic behavior of solar cells. Among these, some are commercially available software such as TCAD, Silvaco Atlas, COMSOL, etc., while few are open sources such as SCAPS-1D, gpvdm, AFORS-HET, PC-1D, AMPS-1D, etc. All of this software fundamentally uses the same semiconductors device models but with different numerical approaches, parameter libraries, and precisions. For this study, SCAPS-1D is selected as it is open-source software, general-purpose in nature, simple in use, suitable for comprehensive analysis, and highly recommended by many researchers for perovskite solar cells as reflected by literature [25,26,27,28,29].

For efficient photovoltaic response, the perovskite active layer is used as an insulator layer, which should be sandwiched between the electron-transport layer (ETL) as the n-type and the hole-transport layer (HTL) as the p-type. Generally, the structures of solar cells can be classified as conventional (n-i-p) or inverted (p-i-n); both are commonly reported structures for perovskite solar cells and have their advantages and disadvantages. In this study, a conventional n-i-p structure is used, where the transparent ETL thin-film is deposited first over the Indium-doped tin oxide (ITO). For ETL, the Niobium pentoxide (*Nb_2_O_5_*) compound material is selected due to many reasons, for instance: (i) it is highly suitable for perovskite solar cells, (ii) it is an excellent transparent material, (iii) it is very effective to block the hole injection, (iv) it offers reduced surface recombination compared to other ETL, and (v) it has a wide optical energy bandgap (vi) which can be tunable due to stoichiometry-dependent nature of its energy bandgap [30,31,32]. Similarly, for the HTL, a Poly(3,4-ethylene dioxythiophene)-poly(styrene sulfonate) (*PEDOT:PSS*) polymer material is selected. The *PEDOT:PSS* is a frequently reported hole-transport layer for solar cells that offers several unique and unmatched advantages, such as (i) low cost, (ii) lightweight, (iii) low-temperature processing, (iv) solution processing, (v) superior mechanically flexible and stretchable, (vi) excellent electrical conductivity with proper doping, (vii) a high degree of optical transparency, and (viii) it shows good chemical and mechanical stability [33,34,35].

In this modeling and simulation study, little attention is paid to the device fabrication process and technology. Despite the fewer successful efforts reported by Liu, Kaushik, and others, the deposition of a *PEDOT:PSS* thin film over the perovskite layer is still a challenging task [36,37,38].

Figure 1 shows the device structure of the three proposed solar cell devices (a) *Nb_2_O_5_/Cs_2_TiI_6_/PEDOT:PSS*, (b) *Nb_2_O_5_/Cs_2_TiBr_6_/PEDOT:PSS*, and (c) *Nb_2_O_5_/Cs_2_TiCl_6_/PEDOT:PSS*, which are used in this research study. This figure also shows the energy bandgap diagram for each proposed device, and it can be noticed that the LUMO level (Lowest Unoccupied Molecular Orbital, conduction band) and HOMO (Highest Occupied Molecular Orbital, valence band) of both *Nb_2_O_5_* and *PEDOT:PSS* layers are compatible with the *Cs_2_TiX_6_* family and are sufficient to support electron-transport and blocking of the hole for *Nb_2_O_5_* as ETL and vice versa for *PEDOT:PSS* as HTL, respectively, which is an essential criterion for the selection of both *Nb_2_O_5_* and *PDOT:PSS* materials as electron and hole-transport layers, respectively, for **Cs_2_TiX_6_*-*based perovskite solar cells.

The main objective of this study is to explore the highly efficient family member of the *Cs_2_TiX_6_* absorber in conjunction with *Nb_2_O_5_* and *PEDOT:PSS* software for perovskite solar cells. For this purpose, these proposed solar cells are optimized using a series of simulations using SCAPS-1D. At first, both *PEDOT:PSS* and *Nb_2_O_5_* are optimized for their thickness and doping density, and then the thickness of the *Cs_2_TiX_6_* active layer is optimized for each family member of the *Cs_2_TiX_6_* material under study (*Cs_2_TiI_6_*, *Cs_2_TiBr_6_*, and *Cs_2_TiCl_6_*). After that, these devices are further optimized and investigated with numerous iterations to determine which converged device will have the highest photovoltaic responses in terms of open-circuit voltage, short-circuit current, fill factor, and, hence, power-conversion efficiency.

## 2. Device Modeling and Simulation Methods

### 2.1. Simulation Methodology

As discussed above, SCAPS-1D software with the highly stable version (3.3.07, created by Marc Bargeman and his co-workers at University of Gent, Ghent, Belgium) is used for this investigation, since it has been used for other similar simulations [25,26,27]. The software uses standard numerical techniques to execute semiconductor differential equations to model the photovoltaic response of any solar cell, just like many other simulation software. These equations can be classified as follows:(*i*)*Poisson Equation:*

This equation defines the fundamental behavior of electrical potential (ϕ) as a function of various types of electrical charges and their distribution inside the solar cell. In this equation, q is referred to as the electric charge constant that is equivalent to 1.602 × 10^−19^ C, while $\epsilon_{o}$ is the absolute dielectric constant, $\epsilon_{r}$ is the relative dielectric constant of each layer’s material, $N_{D}/N_{A}\;$ is the donor/acceptor doping density, $\rho_{p}/\rho_{n}$ is the hole/electron density distribution, and *p*(*x*)/*n*(*x*) is the hole/electron density distribution as a function of the thickness (*x*). The Poisson equation can be defined as:(1)$$\frac{d^{2}\phi \left(x\right)}{dx^{2}}=\frac{q}{\epsilon_{o}\epsilon_{r}}\;\left(p\left(x\right)-n\left(x\right)+N_{D}-N_{A}+\rho_{p}-\rho_{n}\right)$$

(*ii*)
*Continuity Equations:*


The continuity equations give the relation of the rate of electron current density ($J_{n}$) and the rate of hole current density ($J_{p}$) as a function of position (*x*) in terms of carrier generation (*G*), and carrier recombination (*R*). Continuity equations can be described as:(2)$$\frac{dJ_{n}}{dx}=G-R$$
(3)$$\frac{dJ_{p}}{dx}=G-R$$

(*iii*)
*Charge Transport Equations:*


A solar cell is inherently a diode in nature; therefore, the diode charge transport equations can also be applied. These equations combine the total drift and diffusion current of holes with electrons. Here, $\mu_{n}/\mu_{p}$ is the mobility of electron and hole, and $D_{n}/D_{p}$ is the electron/hole diffusion coefficient. The diode charge transport equations can be written as:(4)$$J=J_{n}+J_{p}$$
(5)$$J_{n}=D_{n}\;\frac{dn}{dx}+\mu_{n\;}n\frac{d\phi }{dx}$$
(6)$$J_{p}=-D_{p}\;\frac{dp}{dx}+\mu_{p\;}p\frac{d\phi }{dx}$$


*(iv) *
*Absorption Coefficient Equation:*


The optical absorption coefficient α (λ) is another important parameter, and it can be defined as the average penetration distance covered by a series of photons at a given wavelength (λ) flowing into the semiconductor layers (with a defined bandgap $E_{g}$) before their absorption. Due to the variety of semiconducting materials, different models for the estimation of the absorption coefficient are available in SCAPS-1D software. Here, we employed the most reported optical absorption coefficient model for perovskite solar cells, which can be defined mathematically as in Equation (7),
(7)$$\alpha \;(\lambda )=\left(A+\frac{B}{h\nu }\right)\;\sqrt{h\nu -E_{g}}$$
where *A* and *B* are arbitrary constants, *h* is Plank’s constant, ν is the optical frequency of photons, and $E_{g}$ is the energy band gap of the absorber layer.

### 2.2. Simulation Parameters

The required material and physical parameters for the *PEDOT:PSS*, *Nb_2_O_5_*, *Cs_2_TiCl_6_*, *Cs_2_TiBr_6_*, and *Cs_2_TiI_6_* semiconducting layers used in the simulation are listed in Table 1. Special attention was paid to the selection of these parameters, which were extracted from well-reputed literature, and hence references are provided in the last row of the table. As these materials are inherently disordered semiconductors in nature and have some defects in their bulk region, the presence of these bulk defects, especially for the photo absorber perovskite layer, can significantly harm the overall photovoltaic responses of the solar cell [39,40,41,42]. Therefore, a neutral defects density of 10^14^ cm^−3^, which is distributed uniformly with a 10^15^ cm^2^ electron and hole capture cross-section, are introduced into the bulk region of each semiconducting layer, including *PEDOT:PSS*, as listed in Table 1. *PEDOT:PSS* is a conducting polymer, which is inherently different in nature compared to inorganic semiconductors, especially concerning the distribution of bulk and interface defects. Therefore, to model a realistic photovoltaic response of *PEDOT:PSS*, an exponential distribution of defects with various other parameters is introduced in the simulation. A list of these defect parameters is presented in Table 2. Finally, all simulation studies were executed at a 300 K room temperature environment, and a standard illumination of 100 mW/cm^−2^ was used for all photovoltaic measurements through an A.M 1.5 solar simulator.

### 2.3. Simulation Flowchart

Detailed simulation steps followed in this study are illustrated as a flow chart and shown in Figure 2. In the first step of simulation setup, all material, physical, and geometrical parameters (see Table 1 and Table 2) of each layer of the proposed devices are established in the software. To initialize the simulation, various essential operating parameters are adjusted, and then initial random values are reset concerning the film thickness and the doping density. In the second step of the simulation, the *PEDOT:PSS* thickness is optimized for each proposed device through a series of simulations and then updated with the optimized value in the software for further simulations. In the third step of the simulation, the doping density of the *PEDOT:PSS* is optimized and updated before the next simulation. In the next two steps, both the thickness and doping density of *Nb_2_O_5_* as an electron-transport layer for each device are optimized and then updated. In the last stage of the simulation, the thickness of the absorption layer is optimized independently in each of the devices *Cs_2_TiCl_6_*, *Cs_2_TiBr_6_*, and *Cs_2_TiI_6_*, and then these devices are further optimized and investigated with numerous iterations to determine the converged device, which has the highest photovoltaic response in terms of open-circuit voltage, short-circuit current, fill factor, and, hence, power-conversion efficiency.

## 3. Results and Discussion

### 3.1. Optimization of Thickness of PEDOT:PSS Hole-Transport Layer

The thickness optimization of *PEDOT:PSS* is a challenging task due to the various roles being played by the *PEDOT:PSS* at a time. These roles can be outlined as follows: (i) holes extraction from the active layer, (ii) blocking of electrons from the active layer, (iii) pacifying the interface traps density, and (iv) providing an interface to the anode of the device [51,52,53]. Therefore, to determine the optimized thickness of the *PEDOT:PSS* thin film for each device, the photovoltaic responses need to be estimated. This includes measuring the open-circuit voltage, the short-circuit current, the fill factor, and the power-conversion efficiency as a function of the *PEDOT:PSS* thickness. Here, photovoltaic responses were simulated and estimated against the thickness of the *PEDOT:PSS* variation from 20 nm to 150 nm, and the resulting response curves are displayed in Figure 3.

From the plots in Figure 3, it is observed that each device behaves with very similar photovoltaic trends as a function of *PEDOT:PSS* film thickness but at different scales. The open-circuit voltages of all devices are initially slightly increased and then settled as a function of the increasing thickness of *PEDOT:PSS*, whereas the short-circuit currents of all devices are initially increased until reaching the maximum value and then decrease as a function of the increasing thickness of *PEDOT:PSS*. Similarly, fill-factor responses behave oppositely compared to the short-circuit currents; these responses are initially decreased until reaching the minimum value and then increased as a function of the increasing thickness of *PEDOT:PSS*. While power-conversion efficiency responses of all devices behave very similar to the open-circuit responses, which are slightly increased and then become constant as a function of the thickness of *PEDOT:PSS* is increased. As power-conversion efficiency is a decisive parameter, therefore, it can be inferred that 75 nm, 60 nm, and 50 nm are the optimum thicknesses of *PEDOT:PSS* for *Cs_2_TiI_6_*_,_
*Cs_2_TiCl_6_* and *Cs_2_TiBr_6_* devices, respectively.

### 3.2. Optimization of Doping Density of PEDOT:PSS Hole Transport Layer

The high conductive thin film *PEDOT: PSS* being employed as a hole-transport layer plays a fundamental role to increase the charge transport process, as well as improving the overall electrical interconnection for high-efficiency photovoltaic devices and other electronic devices [54,55,56,57]. It is well-reported that the overall conductivity of the *PEDOT:PSS* material can be further improved by various methods of doping such as thermal treatment [58], solvent (organic, ionic, inorganic) treatments [59,60], surfactant treatments [61], and acidic treatments [62]. Especially, sulfuric acid (*H_2_SO_4_*) treatment for doping can significantly enhance the conductivity up to the fourth order for *PEDOT:PSS* [63,64].

Nearly all photovoltaic parameters, either directly or indirectly, are interconnected and can be improved with proper doping of the hole-transport layer. Such improvements in the photovoltaic response can be associated with the enhancement of the conductivity in the hole-transport layer, which in turn further decreases the series resistance. Moreover, higher conductivity in the hole-transport layer enhances the internal electric field distribution, as well as the built-in potential at the metal–polymer interface for the device that supports the hole extraction and blocking of electrons for efficient photovoltaic response [65]. However, on the other hand, very high doping of the hole-transport layer can decrease the shunt resistance and degrade the short-circuit current; therefore, optimization of the doping density of the hole-transport layer is crucial for efficient photovoltaic responses.

To determine the optimized doping density of *PEDOT:PSS* thin film for each device, the photovoltaic responses, as mentioned above, need to be calculated as a function of the *PEDOT:PSS* doping density. In this study, the photovoltaic responses were simulated and calculated against the doping density of the *PEDOT:PSS* variation from 10^11^ cm^−3^ to 10^20^ cm^−3,^ and the resulting response curves are illustrated in Figure 4. From these results, it is observed that open-circuit voltage, fill-factor, and power-conversion efficiency responses of all devices increase with increasing the doping density of the *PEDOT:PSS* hole-transport layer, but different trends are observed for each device. On the other hand, with the increase in the doping density of *PEDOT:PSS*, the short-circuit current for each device increases up to around 10^15^–10^16^ cm^−3^ and then starts to decrease beyond that, which may be due to the reduction in mobility in the presence of higher dopant density scattering effects. As power-conversion efficiency is the decisive parameter, therefore, it is observed that 10^20^ cm^−3^ is the optimum doping density for each device.

### 3.3. Optimization of Thickness of the Nb_2_O_5_ Electron-Transport Layer

As introduced before, the *Nb_2_O_5_* material is selected as the electron-transport layer for the proposed devices. The importance of the thickness optimization for the electron-transport layer is very similar to the hole-transport layer for the photovoltaic responses, which has already been discussed in Section 3.1. Therefore, to determine the optimized thickness of *Nb_2_O_5_* for each device, the photovoltaic responses including open-circuit voltage, short-circuit current, fill-factor, and power-conversion efficiency need to be determined as a function of the *Nb_2_O_5_* thickness. In this study, the photovoltaic responses were simulated and measured against the thickness of the *Nb_2_O_5_* variation from 20 nm to 150 nm, and the resulting response curves are illustrated in Figure 5.

Generally, organic and polymer semiconductors offer low conductivity, mobility, and other charge transport parameters compared to inorganic semiconductors [66,67], therefore, it can be justified that the main bottleneck is *PEDOT:PSS* as a hole-transport layer, not by *Nb_2_O_5_* as electron-transport layer [68,69]. The *PEDOT:PSS* was previously optimized for all devices, which is required to balance the transport of holes and electrons from respective transport layers for the proposed solar devices. Therefore, the results shown in Figure 5 demonstrate that all devices have similar photovoltaic trends but with different magnitudes, and with very close observations, it can be realized that these photovoltaic responses are slowly decreased with the increase in *Nb_2_O_5_* thickness. Hence, it can be deduced that 20 nm is the optimum thickness of the *Nb_2_O_5_* electron-transport layer for all the proposed devices.

### 3.4. Optimization of Doping Density of the Nb_2_O_5_ Electron-Transport Layer

Figure 6 presents the photovoltaic simulation results against the doping density of the *Nb_2_O_5_* variation from 10^11^ cm^−3^ to 10^20^ cm^−3^, which is used to determine the optimized doping density of the *Nb_2_O_5_* electron-transport layer for each device.

The doping density optimization of the *Nb_2_O_5_* electron-transport layer is as important as the doping density optimization of the *PEDOT:PSS* hole-transport layer. However, in some respects, the electron-transport layer is thought to be more crucial compared to the hole-transport layer, considering that it interacts directly with photons before reaching the active layer. As discussed before, the charge transport parameters of *Nb_2_O_5_* are generally better than *PEDOT:PSS* [68,69]. Therefore, in the presence of the optimized *PEDOT:PSS*, the photovoltaic response parameters of all devices improve with an increasing doping density of *Nb_2_O_5_*, and these improvements are more prominent for the *Cs_2_TiI_6_* device. Looking at the power-conversion efficiency responses, it can be observed that 10^20^ cm^−3^ is the optimum doping density of *Nb_2_O_5_* for each device.

### 3.5. Optimization of Thickness of the Active Perovskite Layer

The thickness optimization of the active perovskite layer is another crucial parameter to define the overall photovoltaic response of the proposed solar devices. Unlike the active polymer layer, the thin perovskite layer has poor optical absorption and, hence, fewer electron-holes will be generated. Similarly, a very thick layer (above then few microns) of perovskite severely degrades the overall photovoltaic responses. It is well-reported that the photovoltaic performances of various perovskite thin films are degraded when their thickness increases from 1000 nm [70]. Such degradation behavior may be attributed to the additional charge recombination with low mobility, which in turn leads to poor photovoltaic efficiency in perovskite solar cells [71]. Therefore, the optimized thickness of active perovskite layers (*Cs_2_TiI_6_*, *Cs_2_TiBr_6_*, and *Cs_2_TiCl_6_*) was estimated by simulating the photovoltaic responses, including open-circuit voltage, short-circuit current, fill-factor, and power-conversion efficiency, as a function of the thickness of the active perovskite layer varying from 75 nm to 1000 nm, and the results are recorded, plotted, and displayed in Figure 7.

The resulting plots demonstrate the complex behavior of photovoltaic response parameters for each device as a function of active layer thickness. As our main objective is to determine the optimal thickness of the active layer, therefore, the thickness associated with maximum power-conversion efficiency is used to estimate the optimal thickness of the active layer. From a thickness point of view, all devices *Cs_2_TiI_6_*, *Cs_2_TiBr_6_*, and *Cs_2_TiCl_6_* have similar power-conversion efficiency responses and reached maximum at 700 nm, 500 nm, and 800 nm thickness, respectively, and then these responses start to decrease at different rates. Therefore, it can be justified that the optimized thickness of each active layer is the resultant optimization outcome of various background factors such as optical absorption, charge transport process, and leakage current at their respective interfaces for each proposed device. Finally, all the optimum parameters concerning thickness and doping density for each layer for the given proposed devices are listed in Table 3.

### 3.6. Overall Photovoltaic Response of the Proposed Devices

Finally, the optimized devices with different active perovskite layers are used to determine the overall photovoltaic response, as shown in Figure 8. It can be realized from this figure that fully optimized devices for *Cs_2_TiI_6_*, *Cs_2_TiBr_6_*, and *Cs_2_Til_6_* offer photovoltaic responses from the range of 10% to 18%, approximately, where *Cs_2_Til_6_* offers the best performance as compared to the other photovoltaic devices. Detailed information about the resultant photovoltaic parameters for each device is listed in Table 4.

Very interesting results are observed for the proposed photovoltaic devices. The device containing *Cs_2_TiCl_6_* shows the highest open-circuit voltage of 1.477 volts but has the lowest short-circuit current of 9.98 mA.cm^−2^. Similarly, the device containing *Cs_2_TiI_6_* offers a maximum short-circuit current of 17.796 and a maximum fill factor of 75%, with a mediocre open-circuit voltage, in addition to up to 18.53% of power-conversion efficiency, which is the best power-conversion efficiency compared to the other devices.

The reason why *Cs_2_TiCl_6_* did not perform well (lowest efficiency, 10.03%) with the highest open-circuit voltage is very straightforward. Among other perovskites, active layer *Cs_2_TiCl_6_* has the highest energy band gap (2.23 eV), whereas *Cs_2_TiI_6_* and *Cs_2_TiBr_6_* both have an energy bandgap of 1.8 eV. The open-circuit voltage ($V_{OC}$) is directly proportional to the energy bandgap ($E_{g}$) of the absorber layer, as in [72,73].
(8)$$V_{OC}=\frac{E_{g}}{q}-\frac{kT}{q}\ln\left(\frac{J_{SC}}{J_{oo}}\right)$$
where $J_{SC}$ is the short-circuit current, and $J_{oo}$ is the current pre-factor for minority carriers. Since all devices are the same except the active perovskite layer, therefore, such high open-circuit voltage may be attributed to the higher energy bandgap of the *Cs_2_TiCl_6_* perovskite layer-based device. On the other hand, a comparatively very high band gap of *Cs_2_TiCl_6_* worsens the photon absorption and leads to the poor generation of the free electron-hole pair and, hence, offers lower short-circuit current and vice versa for *Cs_2_TiI_6_* and *Cs_2_TiBr_6_* with a lower energy band gap, as can be observed from Table 4. Finally, the best performance exposed by *Cs_2_TiI_6_* may be attributed to the better electronic structure and band alignment compared to other perovskite absorber layers, as can be observed from Table 1.

## 4. Conclusions

Finally, it can be concluded that despite having the most efficient photovoltaic response, lead-based halide perovskite solar cells are facing some serious concerns regarding lead-based toxicity. Therefore, many lead-free perovskites have been reported, among these, the cesium titanium halide family (*Cs_2_TiX_6_*, *X* = *Cl*, *Br*, *I*) is becoming very popular. The main objective of this study is to design and compare an efficient cesium titanium halide-based solar cell that can be used as an alternative to replacing lead-based perovskite solar cells. For efficient photovoltaic requirements, the hole-transport layer and electron-transport layer materials such as *PEDOT:PSS* and *Nb_2_O_5_* are selected, as these are the commonly reported materials and are electronically compatible with the cesium titanium halide family. Similarly, for the active perovskite layer, the cesium titanium halide family members such as *Cs_2_TiCl_6_*, *Cs_2_TiBr_6_*, and *Cs_2_TiI_6_* are selected for the proposed perovskite solar cell devices such as *ITO/Nb_2_O_5_/Cs_2_TiI_6_/PEDOT:PSS/Au*, *ITO/Nb_2_O_5_/Cs_2_TiBr_6_/PEDOT:PSS/Au*, and *ITO/Nb_2_O_5_/Cs_2_TiCl_6_/PEDOT:PSS/Au*, respectively. For each proposed device, all layers (*PEDOT:PSS*, *Nb_2_O_5_*, and active absorber perovskite layer) of the device were individually optimized, and then each optimized device was systematically investigated for its photovoltaic responses through simulation to determine the most efficient solar perovskite device. It is observed that the device *ITO/Nb_2_O_5_/Cs_2_TiCl_6_/PEDOT:PSS/Au* displayed the worst performance for power-conversion efficiency (10%), which may be due to the higher energy band gap leading to inefficient optical absorption, while the device *ITO/Nb_2_O_5_/Cs_2_TiI_6_/PEDOT:PSS/Au* showed the most efficient photovoltaic response in terms of power-conversion efficiency of 18.53%, which may be attributed to the better electronic structure and band alignment compared to the other proposed devices.

## Figures and Tables

**Figure 1 nanomaterials-12-03435-f001:**
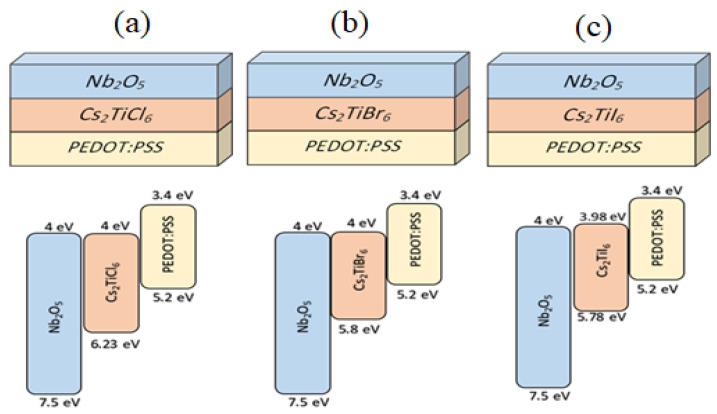
The devices’ structures and the energy band diagram for the proposed perovskite solar cell devices as: (**a**) *Nb_2_O_5_/Cs_2_TiCl_6_/PEDOT:PSS*, (**b**) *Nb_2_O_5_/Cs_2_TiBr_6_/PEDOT:PSS*, and (**c**) *Nb_2_O_5_/Cs_2_TiI_6_/PEDOT:PSS*.

**Figure 2 nanomaterials-12-03435-f002:**
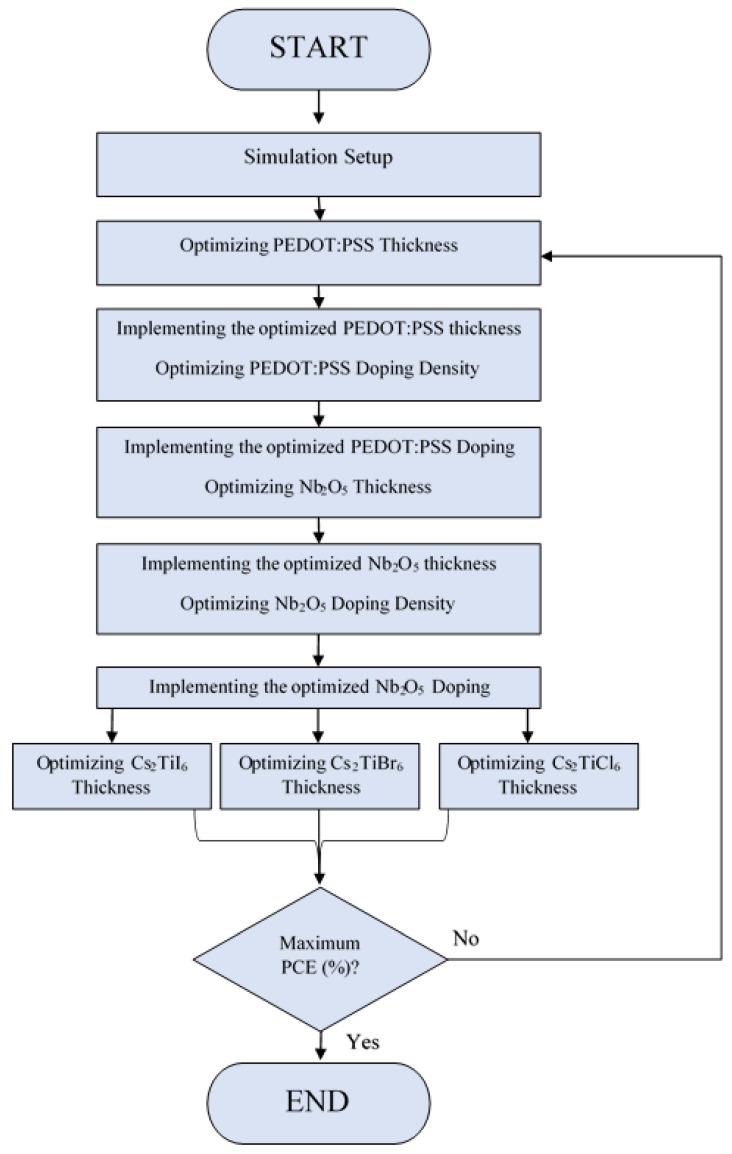
The flow chart of the process used to determine the maximum power-conversion efficiency of the proposed devices *Cs_2_TiCl_6_*, *Cs_2_TiBr_6_*, and *Cs_2_TiI_6_* respectively.

**Figure 3 nanomaterials-12-03435-f003:**
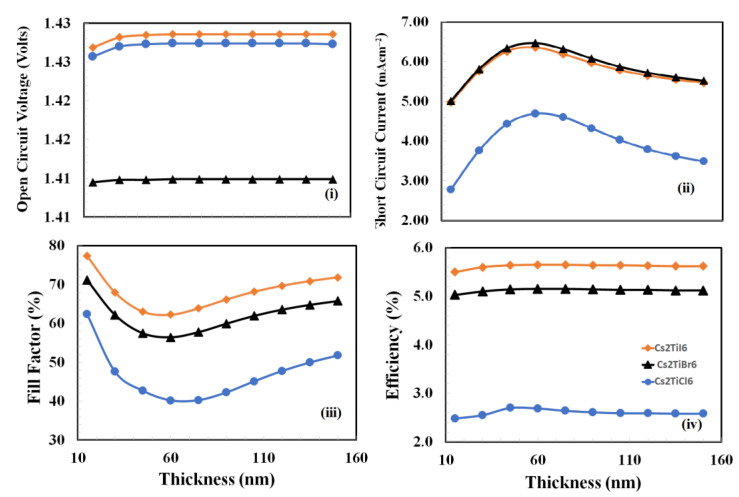
The photovoltaic response parameters: (**i**) open-circuit voltage, (**ii**) short-circuit current, (**iii**) fill-factor, and (**iv**) power-conversion efficiency, as a function of the thickness of *PEDOT:PSS* thin film (hole-transport layer) for the proposed devices *Cs_2_TiCl_6_*, *Cs_2_TiBr_6_*, and *Cs_2_TiI_6_* respectively.

**Figure 4 nanomaterials-12-03435-f004:**
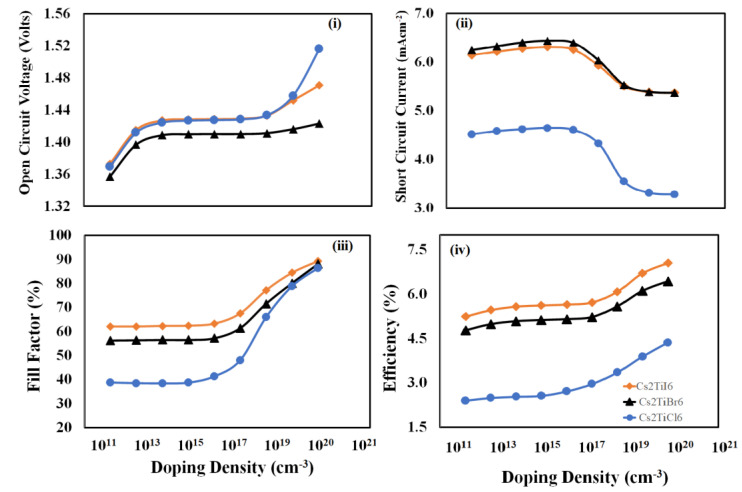
The photovoltaic response parameters: (**i**) open-circuit voltage, (**ii**) short-circuit current, (**iii**) fill-factor, and (**iv**) power-conversion efficiency as a function of the doping-density of *PEDOT:PSS* thin film (hole-transport layer) for the proposed devices *Cs_2_TiCl_6_*, *Cs_2_TiBr_6_*, and *Cs_2_TiI_6_* respectively.

**Figure 5 nanomaterials-12-03435-f005:**
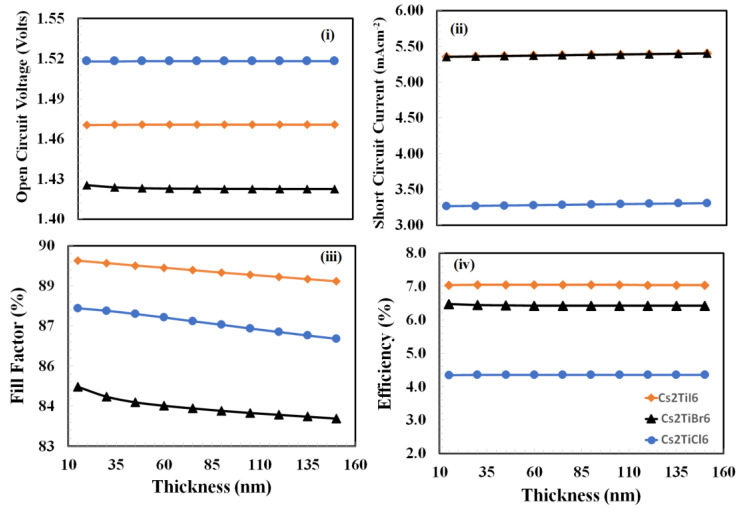
The photovoltaic response parameters: (**i**) open-circuit voltage, (**ii**) short-circuit current, (**iii**) fill-factor, and (**iv**) power-conversion efficiency as a function of the thickness of *Nb_2_O_5_* (electron-transport layer) for the proposed devices *Cs_2_TiCl_6_*, *Cs_2_TiBr_6_*, and *Cs_2_TiI_6_* respectively.

**Figure 6 nanomaterials-12-03435-f006:**
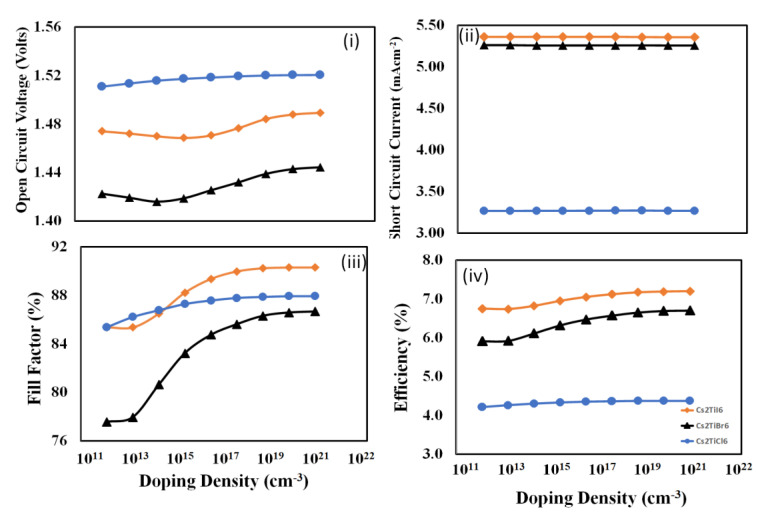
The photovoltaic response parameters: (**i**) open-circuit voltage, (**ii**) short-circuit current, (**iii**) fill-factor, and (**iv**) power-conversion efficiency as a function of the doping density of *Nb_2_O_5_* (electron-transport layer) for the proposed devices *Cs_2_TiCl_6_*, *Cs_2_TiBr_6_*, and *Cs_2_TiI_6_* respectively.

**Figure 7 nanomaterials-12-03435-f007:**
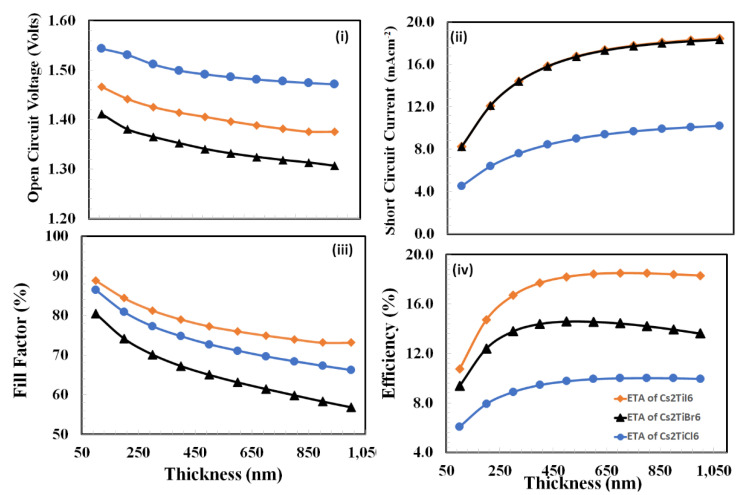
The photovoltaic response parameters: (**i**) open-circuit voltage, (**ii**) short-circuit current, (**iii**) fill-factor, and (**iv**) power-conversion efficiency as a function of the thickness of the active perovskite layer for the proposed devices *Cs_2_TiCl_6_*, *Cs_2_TiBr_6_*, and *Cs_2_TiI_6_* respectively.

**Figure 8 nanomaterials-12-03435-f008:**
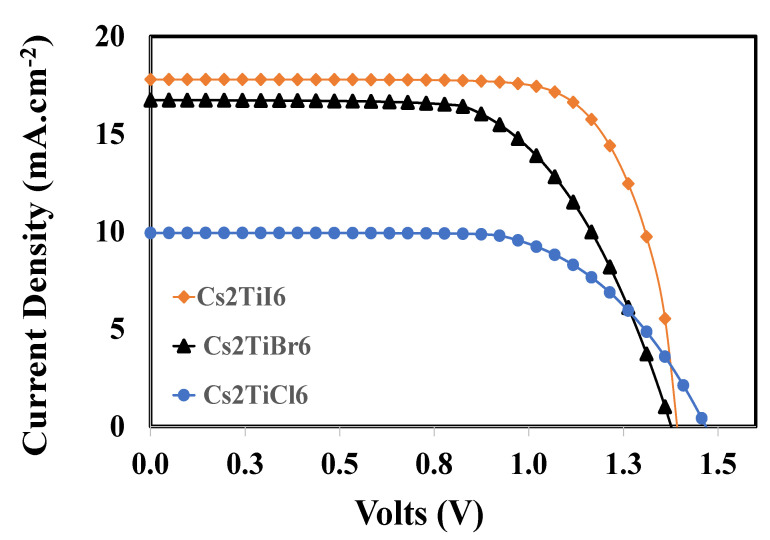
The overall photovoltaic response of the optimized solar perovskite devices.

**Table 1 nanomaterials-12-03435-t001:** The physical and material parameters for active perovskite layers, a hole-transport layer, and an electron-transport layer are considered in the simulation by SCAPS 1-D.

Photovoltaic Parameters	Symbol	Unit	*PEDOT:PSS*	*Nb_2_O_5_*	*Cs_2_TiCl_6_*	*Cs_2_TiBr_6_*	*Cs_2_TiI_6_*
Thickness	*Th*	nm	50	50	50	50	50
Energy Band Gap	$E_{g}$	eV	1.8	3.5	2.23	1.8	1.8
Electron Affinity	*Χ*	eV	3.4	4	4	4	3.98
Dielectric Permittivity	$\epsilon_{r}$		18	9	19	10	7.3
Effective Density of States at Conduction Band	$N_{c}$	cm^−3^	2.2 × 10^19^	2.2 × 10^18^	1 × 10^19^	6 × 10^19^	1 × 10^19^
Effective Density of States at Valence Band	$N_{v}$	cm^−3^	1.8 × 10^19^	1.8 × 10^19^	1 10^19^	2.14 × 10^19^	1 × 10^18^
Hole Thermal Velocity	$V_{h}$	cm/s	1 × 10^7^	1 × 10^7^	1 × 10^7^	1 × 10^7^	1 × 10^7^
Electron Thermal Velocity	$V_{e}$	cm/s	1 × 10^7^	1 × 10^7^	1 × 10^7^	1 × 10^7^	1 × 10^7^
Electron Mobility	$\mu_{e}$	cm^−2^/V·s	4.5 × 10^-02^	25	4.4	2.36 × 10^-01^	4.4
Hole Mobility	$\mu_{h}$	cm^−2^/V·s	4.5 × 10^-02^	5	2.5	1.7 × 10^-01^	2.5
Uniform Shallow Donor Doping	$N_{d}$	cm^−3^	-	1 × 10^16^	1 × 10^19^	1 × 10^19^	1 × 10^19^
Uniform Shallow Acceptor Doping	$N_{a}$	cm^−3^	1 × 10^16^	-	1 × 10^19^	1 × 10^19^	1 × 10^19^
Defect Density	$N_{t}$	cm^−3^	1 × 10^15^	1 × 10^14^	1 × 10^14^	1 × 10^14^	1 × 10^14^
Reference			[43]	[44]	[45,46]	[47]	[48,49,50]

**Table 2 nanomaterials-12-03435-t002:** The defect parameters for active perovskite layers, a hole-transport layer, and an electron-transport layer are considered in the simulation by SCAPS 1-D.

Parameters	Unit	*PEDOT:PSS*	*Nb_2_O_5_*	*Cs_2_TiCl_6_*	*Cs_2_TiBr_6_*	*Cs_2_TiI_6_*
Defect Types		Neutral	Neutral	Neutral	Neutral	Neutral
Capture Cross-section electrons	cm^2^	1 × 10^−15^	1 × 10^−15^	1 × 10^−15^	1 × 10^−15^	1 × 10^−15^
Capture Cross-section holes	cm^2^	1 × 10^−15^	1 × 10^−15^	1 × 10^−15^	1 × 10^−15^	1 × 10^−15^
Energetic Distribution		Uniform	Single	Single	Single	Single
Energy Level with respect to reference	eV	0.6	0.6	0.6	0.6	0.6
Characteristic Energy	eV	0.1	-	-	-	-
Traps Distribution N(t)		Exponential	Uniform	Uniform	Uniform	Uniform
N(t) Left x=0	cm^−3^	1 × 10^13^	1 × 10^14^	1 × 10^14^	1 × 10^14^	1 × 10^14^
N(t) Right x = 1	cm^−3^	1 × 10^15^	-	-	-	-
N(t) Peak x = 0	1/eV/cm^3^	1 × 10^14^	-	-	-	-
N(t) Peak x = 1	1/eV/cm^3^	1 × 10^16^	-	-	-	-

**Table 3 nanomaterials-12-03435-t003:** The overall optimization parameters (thickness and doping density) of hole-transport, electron-transport, and active perovskite layer for proposed photovoltaic devices.

Photovoltaic Devices	*PEDOT:PSS* (HTL)		*Nb_2_O_5_* (ETL)		Active Perovskite Layer
	Thickness	Doping Density	Thickness	Doping Density	Thickness
	(nm)	(/cm^3^)	(nm)	(/cm^3^)	(nm)
Device 1 (CS_2_TiI_6_)	75	10^20^	20	10^20^	700
Device 2 (CS_2_TiBr_6_)	60	10^20^	20	10^20^	500
Device 3 (CS_2_TiCl_6_)	50	10^20^	20	10^20^	800

**Table 4 nanomaterials-12-03435-t004:** The photovoltaics parameters’ open-circuit voltage, short-circuit current, fill-factor, and power-conversion efficiency of the fully optimized devices (a) *Cs_2_TiI_6_*, (b) *Cs_2_TiBr_6_*, and (c) *Cs_2_TiCl_6_*, respectively.

Devices	Open-Circuit Voltage	Short-Circuit Current	Fill-Factor	Power-Conversion Efficiency
	(volts)	(mA.cm^−2^)	(%)	(%)
(a) Cs_2_TiI_6_	1.388	17.796	75	18.53
(b) Cs_2_TiBr_6_	1.341	16.739	65	14.59
(c) Cs_2_TiCl_6_	1.477	9.98	68	10.03

## Data Availability

Available on request.

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
