# Peer review of "Towards Highly Efficient Cesium Titanium Halide Based Lead-Free Double Perovskites Solar Cell by Optimizing the Interface Layers"

_nanomaterials, 2022, doi:10.3390/nano12193435_

Round 1
Reviewer 1 Report
The article deals with the extremely important subject of modern perovskite solar cells. The article is stylistically and graphically correct, only a slight linguistic correction is required. Nevertheless, before it is recommended for publication, and one significant thing needs to be clarified:
Changing the thickness of one layer affects the optimal thickness of the next. Therefore, choosing the thickness of the PEDOT: PSS layer, assuming the thickness of the active layer, and then optimizing the thickness under it, could not give the optimal effect in the form of solar cell efficiency. After optimizing the thickness of the active layer, did the authors check the optimal thickness of PEDOT: PSS and Nb2O5 in the model again?
After clarifying this aspect, the article can be recommended for publication.
Author Response
Author's Response to The Reviewer 1:
- The article deals with the extremely important subject of modern perovskite solar cells. The article is stylistically and graphically correct, only a slight linguistic correction is required. Nevertheless, before it is recommended for publication, and one significant thing needs to be clarified:
Author’s Response:
- Thank you so much for your kind evaluation. We are grateful for your deep review and valuable comments and especially for “The article is stylistically and graphically correct,”
- Changing the thickness of one layer affects the optimal thickness of the next. Therefore, choosing the thickness of the PEDOT: PSS layer, assuming the thickness of the active layer, and then optimizing the thickness under it, could not give the optimal effect in the form of solar cell efficiency. After optimizing the thickness of the active layer, did the authors check the optimal thickness of PEDOT: PSS and Nb2O5 in the model again?
Author’s Response:
- Thank you so much again for your extremely valuable observation. We unanimously agree to further clarify by introducing some text in the manuscript.
- Page: 4, Line:136 (color Yellow)
- At first, both PEDOT:PSS and Nb2O5 are optimized for their thickness and doping density, and then the thickness of the Cs2TiX6 active layer is optimized for each family member of the Cs2TiX6 material under study (Cs2TiI6, Cs2TiBr6, and Cs2TiCl6). After that, these devices are further optimized and investigated with numerous iterations to determine which converged device will have the highest photovoltaic responses in terms of open-circuit voltage, short-circuit current, fill factor, and hence power-conversion efficiency.

Reviewer 2 Report
Moiz and coauthors present an article about the promising of Cesium Titanium Halide Based Lead-Free Double Perovskites Solar Cell based on scientific calculation and simulation. The main objective of this study is to design and compare an efficient cesium titanium halide-based solar cell that can be used as an alternative to replace lead-based perovskite solar cells. The authors claimed that the device of ITO/Nb2O5/Cs2TiI6/PEDOT:PSS/Au shows the most efficient photovoltaic response with little above 21 % as maximum power-conversion efficiency. Some of the findings in this work may be significant for guiding the research about high-efficient lead free organic solar cells, but some concerns must be addressed before publication. I recommend its publication in Nanomaterials after major revision:
1. About the title, the authors mentioned that is for Cesium Titanium Halide Based Lead-Free Double Perovskites. However, all the simulation and calculation are about how the interlayer (ETL and HTL) effect the devices performances. We know that the major challenging about novel lead-free perovskite solar cells is the phase stability and defects control of the perovskite structure not the interfacial layer. Here if authors only focus on the interlayer and I would suggest that authors change the title to “Towards Highly Efficient Cesium Titanium Halide Based Lead-Free Double Perovskites Solar Cell by modification the interface layers” or something which more related interfacial instead of perovskite layer.
2. I totally understand that is a calculation works but I still think the basic materials choosing should be follow the experiments basis. Here authors choosing the PEDOT:PSS as HTL for n-i-p structure, but usually we don’t use PEDOT:PSS due to the solvent compatibility. It is really hard to spin the water based PEDOT:PSS on perovskite layer since it really ease to decompose by water. Here I am wondering did authors consider about the water resistance of Cs2TiX3 system?
3. About the thickness of PEDOT:PSS, it is strange that when increasing the thickness, Jsc and PCE are all increased, as we know the PEDOT:PSS didn’t has enough conductivity or hole mobility to extract the free hole from active layer, when thickness increased, also resistance will increase. I am pretty sure if we are using 300 nm PEDOT as HTL in real devices, the Jsc should be trend to 0. It didn’t make any sense to me here. I would suggest authors think about the model and also please check some reference based on real devices.
Overall, my suggestion about this work is major revision.
Author Response
Author’s Response:
Thank you so much for your kind evaluation and efforts to improve the quality of the manuscript. We all are grateful from the bottom of our hearts for your deep review and valuable suggestions.
- About the title, the authors mentioned that is for Cesium Titanium Halide Based Lead-Free Double Perovskites. However, all the simulation and calculation are about how the interlayer (ETL and HTL) effect the devices performances. We know that the major challenging about novel lead-free perovskite solar cells is the phase stability and defects control of the perovskite structure not the interfacial layer. Here if authors only focus on the interlayer and I would suggest that authors change the title to “Towards Highly Efficient Cesium Titanium Halide Based Lead-Free Double Perovskites Solar Cell by modification the interface layers” or something which more related interfacial instead of perovskite layer.
Author’s Response:
- Thank you so much again for your extremely valuable observation. We unanimously agree to modify the title of the manuscript according to the instructions of the reviewer.
- Page: 1, Line:1 (color Yellow)
- Towards Highly Efficient Cesium Titanium Halide Based Lead-Free Double Perovskites Solar Cell by Optimizing the Interface Layers
- I totally understand that is a calculation works but I still think the basic materials choosing should be follow the experiments basis. Here authors choosing the PEDOT:PSS as HTL for n-i-p structure, but usually we don’t use PEDOT:PSS due to the solvent compatibility. It is really hard to spin the water based PEDOT:PSS on perovskite layer since it really ease to decompose by water. Here I am wondering did authors consider about the water resistance of Cs2TiX3 system?
Author’s Response:
Thank you so much for your very deep evaluation of the manuscript. We completely agree with the reviewer’s point of view and incorporate the following text in the manuscript in response to the reviewer’s concern.
- Page: 4, Line:114 (color Yellow)
- In this modeling and simulation study, little attention is paid to the device fabrication process and technology. Despite the fewer successful efforts reported by Liu, Kaushik, and others, the deposition of PEDOT:PSS thin film over the perovskite layer is still a challenging task [36-38]
- [36] Koushik, D.; Verhees, W.; Zhang, D.; Kuang, Y.; Veenstra, S.; Creatore, M.; Schropp, R. Atomic Layer Deposition Enabled Perovskite/PEDOT Solar Cells in a Regular n-i-p Architectural Design. Advanced Materials Interfaces 2017, 4, 1700043. [CrossRef]
- [37] Liu, J.; Pathak, S.; Stergiopoulos, T.; Leijtens, T.; Wojciechowski, K.; Schumann, S.; Kausch-Busies, N.; Snaith, H. Employing PEDOT as the p-Type Charge Collection Layer in Regular Organic–Inorganic Perovskite Solar Cells. The Journal of Physical Chemistry Letters 2015, 6, 1666-1673. [CrossRef]
- [38] Hou, Y.; Zhang, H.; Chen, W.; Chen, S.; Quiroz, C.; Azimi, H.; Osvet, A.; Matt, G.; Zeira, E.; Seuring, J.; Kausch-Busies, N.; Lövenich, W.; Brabec, C. Inverted, Environmentally Stable Perovskite Solar Cell with a Novel Low-Cost and Water-Free PEDOT Hole-Extraction Layer. Advanced Energy Materials 2015, 5, 1500543. [CrossRef]
- About the thickness of PEDOT:PSS, it is strange that when increasing the thickness, Jsc and PCE are all increased, as we know the PEDOT:PSS didn’t has enough conductivity or hole mobility to extract the free hole from active layer, when thickness increased, also resistance will increase. I am pretty sure if we are using 300 nm PEDOT as HTL in real devices, the Jsc should be trend to 0. It didn’t make any sense to me here. I would suggest authors think about the model and also please check some reference based on real devices.
Overall, my suggestion about this work is major revision.
Author’s Response:
Thank you so much for your very deep evaluation of the manuscript. We completely agree with the reviewer’s point of view and have made a lot of changes to address the reviewer’s concerns.
- We introduced exponential distribution defects in PEDOT:PSS to be a more realistic simulation and PEDOT:PSS thickness was found close to 60 nm for each device. See Figure 3
- We introduced a new Table 2 for defects parameters introduced in the simulation
- We repeated all simulations with new defects parameters
- We changed all figures according to the new simulation results.
- We modified the given text in the manuscript.
- Page 1, Line: 24 (Abstract)
- It is observed that the device ITO/Nb2O5/Cs2TiI6/PEDOT:PSS/Au shows the most efficient photovoltaic response with little above 18.5 % as maximum power-conversion efficiency.
- Page 5, Line: 201
- As these materials are inherently disordered semiconductors in nature and have some defects in their bulk region. The presence of these bulk defects especially for the photo absorber perovskite layer can significantly harm the overall photovoltaic responses of the solar cell [39-42]. Therefore, neutral defects density of 1014 cm-3 , which is distributed uniformly with 1015 cm2 electron and hole capture cross-section, are introduced into the bulk region of each semiconducting layer including PEDOT:PSS as listed in Table 1. As PEDOT:PSS is a conducting polymer, which is inherently different in nature compared to inorganic semiconductors, especially concerning the distribution of bulk and interface defects. Therefore, to model a realistic photovoltaic response of PEDOT:PSS, an exponential distribution of defects with various other parameters is introduced in the simulation. A list of these defect parameters is presented in Table 2.
- Page 6, Line: 233 (Table 2)
- Table The defects parameters for active perovskite layers, a hole-transport layer, and electron-transport layer are considered in the simulation by SCAPS 1-D.
- Page 8, Line: 261
- From the plots in Figure 3, it is observed that each device behaves with very similar photovoltaic trends as a function of PEDOT:PSS film thickness but at different scales. The open-circuit voltages of all devices are initially slightly increased and then settled as a function of the increasing thickness of PEDOT:PSS, whereas the short-circuit currents of all devices are initially increased until reaching the maximum value, and then decrease, as a function of the increasing thickness of PEDOT:PSS. Similarly, fill-factor responses behave oppositely compared to the short-circuit currents, these responses are initially decreased, until reaching the minimum value, and then increased as a function of the increasing thickness of PEDOT:PSS. While power-conversion efficiency responses of all devices behave very similar to the open-circuit responses, which are slightly increased and then become constant as a function of the thickness of PEDOT:PSS is increased. As power-conversion efficiency is a decisive parameter, therefore it can be inferred that 75 nm, 60 nm, and 50 nm are the optimum thicknesses of PEDOT:PSS for Cs2TiI6, Cs2TiCl6 and Cs2TiBr6 devices respectively.
- Page 8, Figure 3 (changed)
- Page 9, Figure 4 (changed)
- Page 10, Figure 5 (changed)
- Page 11, Figure 6 (changed)
- Page 12, Figure 7 (Changed)
- Pag 12, Line 389
- . From a thickness point of view, all devices Cs2TiI6, Cs2TiBr6, and Cs2TiCl6 have similar power-conversion efficiency responses and reached maximum at 700 nm, 500 nm, and 800 nm thickness respectively, and then these responses start to decrease at different rates. So, it can be justified that the optimized thickness of each active layer is the resultant optimization outcome of various background factors such as optical absorption, charge transport process, and leakage current at their respective interfaces for each proposed device.
- Pag 12, Line 399. Table 3 (Changed)
- Pag 13, Figure 8 (Changed)
- Pag 13, Line 417. Table 4 (Changed)
- Pag 14, Line 417. Table 4 (Changed)
- Pag 15, Line 457 (Conclusion).
It is observed that the device ITO/Nb2O5/Cs2TiCl6/PEDOT: PSS/Au displayed the worst performance for power-conversion efficiency (10%), which may be due to the higher energy band gap leading to inefficient optical absorption. While the device ITO/Nb2O5/Cs2TiI6/PEDOT:PSS/Au showed the most efficient photovoltaic response in terms of power-conversion efficiency of 18.53%, which may be attributed to the better electro

Round 2
Reviewer 2 Report
The authors addressed all concerns I had. I suggest it would be a good shape to publish in Nanomaterials.